# REORIENTING THE FROZEN SPACE: TRAINING-FREE TEST-TIME ADAPTATION BY GEOMETRIC TRANSFORMATION

## ABSTRACT

With the widespread application of Vision-Language Models (VLMs) in downstream tasks, test-time adaptation methods based on VLMs, particularly the training-free paradigm, have been gaining increasing attention due to their advantages in handling distribution shifts during testing. Yet, existing training-free methods remain constrained by the fixed geometry of pretrained feature spaces, which limits class separability. We propose SOBA, a training-free TTA method that edits decision geometry by re-expressing class prototypes in a test-induced orthogonal basis. SOBA maintains a lightweight dynamic queue of high-confidence test samples, derives an orthogonal basis via singular value decomposition, and aligns prototypes to the most discriminative directions of the test distribution. This simple adjustment enlarges inter-class margins, sharpens decision boundaries, and improves the recognition of semantically similar categories—without modifying features, prompts, or model parameters. Extensive experiments on multiple benchmarks demonstrate that SOBA achieves state-of-the-art accuracy and superior efficiency compared to both training-free and backprop-based TTA methods.

## 1 INTRODUCTION

Visual-language models (VLMs), such as CLIP Radford et al. (2021) and ALIGN Jia et al. (2021), have garnered significant attention for their strong generalization capabilities. To further enhance their performance on downstream tasks, various tuning methods like prompt tuning Zhou et al. (2022b;a); Khattak et al. (2023) and adapter tuning Zhang et al. (2022b); Gao et al. (2024) have been proposed. However, the reliance of these methods on training data fundamentally hinders their generalization to new domains. Therefore, test-time adaptation (TTA) Shu et al. (2022); Karmanov et al. (2024), which leverages incoming test samples without requiring any manual labels to rapidly adjust to downstream data distributions, holds significant promise for practical applications.

Mainstream TTA methods for VLMs fall into two primary paradigms. The first, prompt-tuning TTA, includes methods like TPT Shu et al. (2022) , DiffTPT Feng et al. (2023), and HisTPT Zhang et al. (2024b) that tune prompts via backpropagation. However, their reliance on computationally expensive optimization contradicts the need for rapid adaptation. The second paradigm, training-free TTA, avoids this overhead, with methods like TDA Karmanov et al. (2024) using a dynamic adapter guided by high-quality test samples. Despite its efficiency and competitive performance, this approach is fundamentally *imprisoned within the original, static feature space of the pre-trained model*.

The fundamental issue with this "frozen feature" is most pronounced for semantically similar classes, where their feature representations can significantly overlap, causing the decision boundaries to become inherently blurred (Fig. 1 (a)). Existing training-free methods like TDA, which operate within this fixed geometry, can at best perform minor adjustments to the decision boundary. However, they are powerless to resolve the fundamental problem of the entangled feature space itself, resulting in very limited performance gains on such difficult classes ( Fig. 1 (c)). This raises a critical question: *can we enhance the separability of the feature space without backpropagation?*

Inspired by these insights, we introduce **S**pace re**O**rienting with **B**asis tr**A**nsformation (SOBA), a training-free TTA that edits the decision geometry via prototype alignment in a test-induced orthogonal basis. Unlike prompt/adapter TTA that adjusts parameters within the original coordinates,

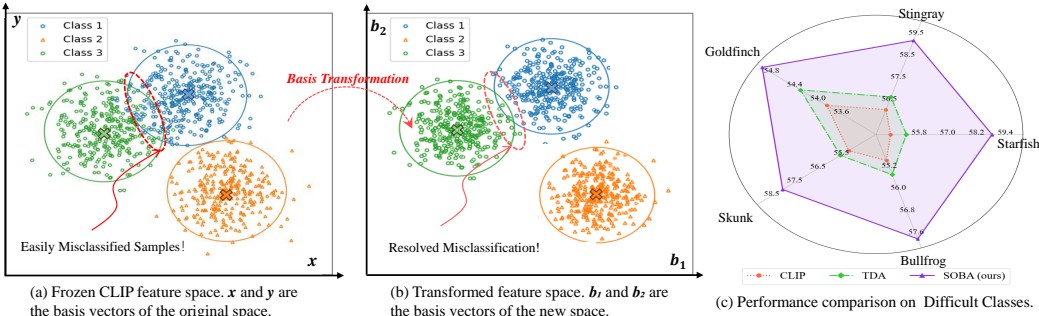

(a) Frozen CLIP feature space. $x$ and $y$ are the basis vectors of the original space.

(b) Transformed feature space. $b_1$ and $b_2$ are the basis vectors of the new space.

(c) Performance comparison on Difficult Classes.

Figure 1: (a) Original CLIP Space. In the original feature space, CLIP may misclassify test samples (red circles) due to overlapping decision boundaries for certain classes. Therefore, for training-free TTA methods, the inability to adjust the feature space limits their applicability in downstream tasks. (b) Feature space after basis transformation. We apply a basis transformation using new vectors (e.g., $b_1$ and $b_2$ in Fig.(b)) to the feature space, making it linearly separable. In this transformed space, we establish clearer decision boundaries, addressing the limitation of training-free TTA methods that cannot adjust the feature space. (c) Performance comparison on the difficult classes. Our method demonstrates a more significant improvement over the current SOTA methods in challenging classes.

SOBA changes the comparison coordinates: prototypes are aligned to the principal directions of the incoming test distribution, enlarging inter-class angular gaps for hard, semantically similar categories—without modifying features or running backprop.

Concretely, SOBA maintains a lightweight dynamic queue of high-confidence pseudo-labeled features, estimates their global structure, and obtains an orthogonal basis via SVD. We then express category means on this data-driven basis, aligning them with directions that concentrate discriminative variation across classes Strang (2000); Abdi & Williams (2010). This alignment enlarges inter-class angular gaps for semantically similar categories—without modifying features, prompts, or weights—and yields clearer boundaries and stronger performance on hard classes Martinez & Kak (2001). As shown in Fig. 1(b), compared with the original CLIP feature space, the transformed space constructed with basis $\mathcal{B}$ exhibits clearer decision boundaries. As illustrated in Fig. 1(c), this enhances the separability of hard classes and leads to notable performance gains.

In this paper, we present three key contributions. First, we identify a key limitation of current training-free TTA methods: their inability to adjust the feature space. To address this bottleneck, we introduce a novel space reorientation method based on basis transformation, which reshapes the feature space to effectively resolve the issue of feature inseparability inherent in the original space. Second, our method achieves state-of-the-art (SOTA) performance across out-of-distribution and cross-dataset benchmarks, effectively adapting to distribution shifts in downstream tasks. Finally, our method also achieves high computational efficiency. Experiments on the ImageNet dataset demonstrate that, compared to the training-free SOTA method TDA Karmanov et al. (2024), our approach achieves a 13.96% speedup in testing and is **56× faster** than the backprop-based method TPT Shu et al. (2022), underscoring its practical applicability.

## 2 RELATED WORKS

**Vision-Language Model.** In recent years, vision-language models have gained widespread attention for their ability to process both visual and linguistic modalities. Models such as CLIP Radford et al. (2021), ALIGN Jia et al. (2021), BLIP Li et al. (2022), and FILIP Yao et al. (2021) leverage self-supervised training on image-text pairs to establish connections between vision and language, enabling strong semantic understanding. This capability allows vision-language models (e.g., CLIP) to exhibit remarkable generalization across various downstream tasks Ding et al. (2022); Maaz et al. (2022); Wang et al. (2024a;b). Prompt tuning and adapter methods have been introduced to enhance the transferability of vision-language models. However, prompt tuning methods (e.g., CoOp Zhou et al. (2022b), CoCoOp Zhou et al. (2022a), Maple Khattak et al. (2023)) and adapter-based approaches (e.g., Tip-Adapter Zhang et al. (2022b), CLIP-Adapter Gao et al. (2024)) typically require large

amounts of training data when adapting to downstream tasks, which limits their applicability in real-world scenarios that demand rapid adaptation. Therefore, this paper focuses on test-time adaptation (TTA) Shu et al. (2022), a method that enables model transfer to downstream tasks without relying on training data.

**Test-Time Adaptation.** TTA enables models to adapt to distribution shifts during inference without training data Boudiaf et al. (2022); Zhang et al. (2022a); Yuan et al. (2023); Zhang et al. (2023); Han et al. (2024); Döbler et al. (2024); Zhang et al. (2024a); Sui et al. (2025); Zhang et al. (2024b). TPT Shu et al. (2022) learns adaptive prompts via entropy minimization, while DiffTPT Feng et al. (2023) enhances diversity using Stable Diffusion Rombach et al. (2022) and filters augmentations by cosine similarity. Both require backpropagation, limiting efficiency. TDA Karmanov et al. (2024) avoids this by leveraging a cache model Zhang et al. (2022b) to refine predictions via test-sample similarity, enabling training-free adaptation. However, it still operates within CLIP's original feature space. We propose mapping features to a spherical space to better handle distribution shifts.

**Statistical Learning.** Statistical learning techniques play an important role in dimensionality reduction and feature extraction. PCA Abdi & Williams (2010), a classic unsupervised method, simplifies data by maximizing its variance, yet its limitation lies in neglecting class information. In contrast, its supervised counterpart, LDA Xanthopoulos et al. (2012), learns more discriminative features by maximizing class separability. Furthermore, SVM Cortes (1995) cleverly employs the kernel trick to achieve effective nonlinear classification in high-dimensional spaces. Inspired by these statistical learning approaches, this paper introduces SOBA, a novel geometric transformation method designed to dynamically reshape the feature space, specifically addressing the problem of train-free TTA.

## 3 METHOD

### 3.1 A TRAINING-FREE BASELINE

The CLIP Radford et al. (2021) model is a pre-trained vision-language model featuring a visual encoder $g_v$ and a text encoder $g_t$. For zero-shot classification with N classes, CLIP leverages these encoders to first obtain text embeddings $\mathbf{W}_t$ from handcrafted class descriptions and a visual embedding $\boldsymbol{f}_{test}$ from a test image $x_{test}$. The model then computes the cosine similarity between $\boldsymbol{f}_{test}$ test and all embeddings in $\mathbf{W}_t$. The final prediction (logits) is then calculated as:

$$logits_{\text{clip}} = \boldsymbol{f}_{test}\mathbf{W}_t^{\text{T}}. \tag{1}$$

As a foundation for our method, we first construct a **training-free baseline** model. Its core involves employing a dynamic queue to store $K$ representative samples for each class and their pseudo-labels $l_p$ for each pseudo-category, where pseudo-labels $l_p$ are generated according to the minimum entropy criterion, specifically by one-hot encoding the prediction scores (Eq 1) for each sample:

$$l_p = \text{OneHot}(\boldsymbol{f}_{test}\mathbf{W}_t^{\text{T}}). \tag{2}$$

Our update strategy for the queue follows the principle of entropy minimization. To be specific, when the queue is full (at its capacity K), we replace the sample exhibiting the highest entropy. We then employ an NCM Mensink et al. (2013b) classifier for the final classification, defined as follows:

$$logits_{\text{ncm}} = \text{sim}(\boldsymbol{f}_{test}, \boldsymbol{\mu}) = \frac{\boldsymbol{f}_{\text{test}} \cdot \boldsymbol{\mu}^T}{\|\boldsymbol{f}_{\text{test}}\| \, \|\boldsymbol{\mu}\|}, \tag{3}$$

where $\text{sim}(\cdot)$ is the cosine similarity, and $\boldsymbol{\mu}$ is the class means for all categories in the queue. This prediction (Eq 3) is then ensembled with that of a zero-shot CLIP (Eq 1) to produce the final inference.

### 3.2 THEORETICAL FOUNDATION

The core of our method lies in reshaping the CLIP feature space via a proposed Geometric Transformation, which aims to overcome the performance bottleneck of existing TTA methods caused by their fixed decision boundaries. This chapter focuses on the theoretical core, orthogonal basis transformation, to establish the mathematical foundation for the subsequent algorithm.

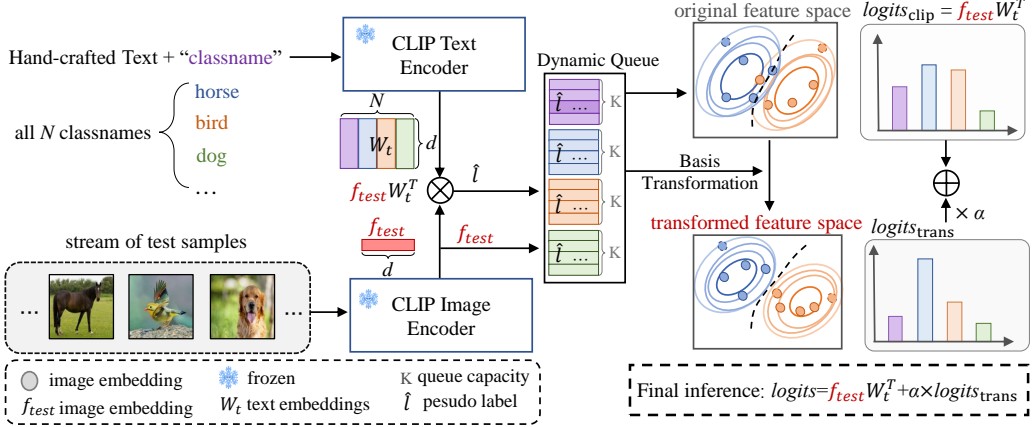

Figure 2: An overview of our method. We maintain a dynamic cache queue of representative samples, selected based on the minimum entropy of CLIP predictions. Using these samples, we construct a basis transformation to reconstruct the feature space and refine decision boundaries. Predictions for test examples are generated through this dynamic mapping and combined with zero-shot CLIP outputs to produce the final inference, with the queue continuously updated during testing.

Given a matrix $\mathbf{V} \in \mathbb{R}^{n \times d}$, it can be expressed as a linear combination of the standard basis matrices $\mathcal{E} = \{\mathbf{e}_{ij} \mid 1 \leq i \leq n, 1 \leq j \leq d\}$. The basis matrix $\mathbf{e}_{ij} \in \mathbb{R}^{n \times d}$ is defined as a matrix whose $(i, j)$-th entry is 1 and all other entries are 0. Thus, the matrix $\mathbf{V}$ can be represented as:

$$\mathbf{V} = \sum_{i=1}^{n} \sum_{j=1}^{d} v_{ij} \mathbf{e}_{ij}, \tag{4}$$

where $v_{ij}$ is the $(i, j)$-th element of $\mathbf{V}$ and serves as its coordinate with respect to the basis matrix $\mathbf{e}_{ij}$. In this paper, our goal is to re-express $\mathbf{V}$ in terms of a different **orthonormal basis** $\mathcal{B} = \{\mathbf{b}_{ij} \in \mathbb{R}^{n \times d}\}_{i \in [n], j \in [d]}$. The vector space $\mathcal{H} := \mathbb{R}^{n \times d}$, equipped with the Frobenius inner product $\langle \mathbf{A}, \mathbf{B} \rangle = \text{trace}(\mathbf{A}^{\mathrm{T}} \mathbf{B})$, forms a finite-dimensional Hilbert space. A fundamental property of such spaces is that an orthonormal basis always exists, and any element $\mathbf{V} \in \mathcal{H}$ can be expanded in terms of any chosen orthonormal basis. For the basis $\mathcal{B}$, this expansion is given by:

$$\mathbf{V} = \sum_{\mathbf{b} \in \mathcal{B}} \langle \mathbf{V}, \mathbf{b} \rangle \, \mathbf{b} = \sum_{i=1}^{n} \sum_{j=1}^{d} \langle \mathbf{V}, \mathbf{b}_{ij} \rangle \, \mathbf{b}_{ij}. \tag{5}$$

We note that the standard basis $\mathcal{E}$ is itself orthonormal. When setting $\mathcal{B} = \mathcal{E}$, Eq. 5 correctly reduces to Eq. 4, since $\langle \mathbf{V}, \mathbf{e}_{ij} \rangle = v_{ij}$.

In summary, Eqs. 4 and 5 describe the process of an orthogonal basis transformation. Since this transformation is structure-preserving, it enables more effective feature representations without distorting intrinsic data relationships Schönemann (1966); Jia et al. (2019). Therefore, it is employed when the standard basis is suboptimal, yielding a more suitable feature space for downstream tasks.

## 3.3 TEST-TIME ADAPTATION BY GEOMETRIC TRANSFORMATION

In this section, we first describe the construction of an appropriate basis for the **SOBA** transformation, and then explain how to implement the transformation by estimating its parameters from the testing data stream.

**Basis Construction** The core of our method is an orthogonal basis transformation. We begin by defining the basis $\mathcal{B} = \{\mathbf{b}_{ij}\}$ using a pair of unitary matrices, $\mathbf{P} \in \mathbb{R}^{n \times n}$ and $\mathbf{Q} \in \mathbb{R}^{d \times d}$. Each basis element is defined as $\mathbf{b}_{ij} := \mathbf{p}_i \mathbf{q}_j^{\mathrm{T}}$, where $\mathbf{p}_i$ and $\mathbf{q}_j$ are the i-th and j-th columns of $\mathbf{P}$ and $\mathbf{Q}$, respectively. As established, this construction ensures $\mathcal{B}$ is an orthonormal basis.

As established by Hilbert space theory (Sec 3.2), the feature matrix $\mathbf{V}$ can be represented in the new coordinate system defined by our basis $\mathcal{B}$. This mapping yields a new coordinate matrix, $\hat{\mathbf{V}}$, which is efficiently computed by the following matrix operation:

$$\hat{\mathbf{V}} = \mathbf{P}^{\mathrm{T}}\mathbf{V}\mathbf{Q}. \tag{6}$$

This operation is the compact form of computing the projection of $\mathbf{V}$ onto every basis element, as defined in our theory (Eq. 5). Specifically, each coordinate $\hat{v}_{ij}$ in $\hat{\mathbf{V}}$ is the result of the inner product:

$$\hat{v}_{ij} = \langle \mathbf{V}, \mathbf{b}_{ij} \rangle = \mathbf{p}_i^{\mathrm{T}}\mathbf{V}\mathbf{q}_j. \tag{7}$$

Conversely, the original matrix $\mathbf{V}$ can be perfectly reconstructed from its new coordinates $\hat{\mathbf{V}}$ ( Detail explanation from Eq. 4 to Eq. 8 refer to Appendix E.3.). This synthesis process is given by:

$$\mathbf{V} = \mathbf{P}\hat{\mathbf{V}}\mathbf{Q}^{\mathrm{T}}. \tag{8}$$

Therefore, the central challenge is *how to select the unitary matrices $\mathbf{P}$ and $\mathbf{Q}$ to make this transformation meaningful* for addressing distribution shifts. Drawing inspiration from PCA theory Abdi & Williams (2010), we leverage the common low-rank property of deep neural network features Aghajanyan et al. (2020). The most informative directions of a feature distribution are its principal components, which are the columns of the unitary matrix $\mathbf{Q}_c$ obtained from the SVD of the feature covariance matrix $\mathbf{C}$:

$$\mathbf{C} = \mathbf{Q}_c\Sigma\mathbf{Q}_c^{\mathrm{T}}. \tag{9}$$

To align our basis with these informative directions, we make a strategic choice: for simplicity, we set $\mathbf{P} = \mathbf{I}_n$ (the identity matrix), and we set $\mathbf{Q} = \mathbf{Q}_c$. This ensures our basis transformation prioritizes the most important structural information within the feature data Aghajanyan et al. (2020).

**Implementation.** We implement the **SOBA** transformation based on the baseline introduced in Sec. 3.1. Specifically, the process begins by computing each class means, $\boldsymbol{\mu} = \{\boldsymbol{\mu}_k\}_1^N$, from the queued features and obtaining a matrix $\boldsymbol{\mu}$ that represents the original decision boundary.

Our objective is to map $\boldsymbol{\mu}$ to a more effective new coordinate space $\hat{\boldsymbol{\mu}}$, using the transformation defined in Eq. 6 to Eq. 9. The core challenge of this process, therefore, is to find the optimal unitary matrix $\mathbf{Q}_c$ that defines this transformation.

According to Eq. 9, we first estimate a shared covariance matrix $\mathbf{C}$ for all classes, following the GDA assumption Hastie & Tibshirani (1996) to reduce computational cost. The estimation requires the class means $\boldsymbol{\mu}_k$, which are empirically estimated from the samples in the dynamic queue:

$$\boldsymbol{\mu}_k = \frac{\sum_{i=1}^{M_k} \mathbb{I}_{\hat{l}=k}\boldsymbol{f}_{test,i}}{\sum_{i=1}^{M_k} \mathbb{I}_{\hat{l}=k}}, \tag{10}$$

where $M_k$ is the number of samples for the pseudo-label class $k$. With the class means, the covariance matrix is then estimated as:

$$\mathbf{C} = \frac{1}{N}\sum_{k=1}^{N} \frac{\sum_{i=1}^{M_k} \mathbb{I}_{\hat{l}=k}(\boldsymbol{f}_{test,i} - \boldsymbol{\mu}_k)(\boldsymbol{f}_{test,i} - \boldsymbol{\mu}_k)^{\mathrm{T}}}{\sum_{i=1}^{M_k}\mathbb{I}_{\hat{l}=k}}. \tag{11}$$

We then perform SVD on $\mathbf{C}$ as shown in Eq. 9 to obtain $\mathbf{Q}_c$.

With our strategic choice of $\mathbf{P} = \mathbf{I}_n$ and $\mathbf{Q} = \mathbf{Q}_c$, the forward transformation (mapping from original to new space) from Eq. 9 simplifies. We apply this simplified mapping to the class means:

$$\hat{\boldsymbol{\mu}} = \boldsymbol{\mu}\mathbf{Q}_c. \tag{12}$$

The transformed logits are subsequently computed by measuring similarity within this new, optimized feature space:

$$logits_{\mathrm{trans}} = \mathrm{sim}(\boldsymbol{f}_{test}, \hat{\boldsymbol{\mu}}). \tag{13}$$

During inference, the statistics $\boldsymbol{\mu}$ and $\mathbf{C}$ are updated periodically (e.g., every 10% of test samples) to adapt to the test distribution while managing computational overhead. The final prediction is a weighted sum of the original CLIP logits (Eq 1) and the transformed logits (Eq 13) :

$$logits = logits_{\mathrm{clip}} + \alpha \cdot logits_{\mathrm{trans}} = \boldsymbol{f}_{test}\mathbf{W}_t^{\mathrm{T}} + \alpha \cdot \mathrm{sim}(\boldsymbol{f}_{test}, \hat{\boldsymbol{\mu}}), \tag{14}$$

| Method | BP | ImageNet | ImageNet-A | ImageNet-V2 | ImageNet-R | ImageNet-S | *Average* | *OOD Average* |
|---|---|---|---|---|---|---|---|---|
| *(a) Full results on the OOD Benchmark with ResNet50 backbone* | | | | | | | | |
| CLIP | ✗ | 59.81 | 23.24 | 52.91 | 60.72 | 35.48 | 46.43 | 43.09 |
| CoOp Zhou et al. (2022b) | ✓ | 63.33 | 23.06 | 55.40 | 56.60 | 34.67 | 46.61 | 42.43 |
| CoCoOp Zhou et al. (2022a) | ✓ | 62.81 | 23.32 | 55.72 | 57.74 | 34.48 | 46.81 | 42.82 |
| Tip-Adapter Zhang et al. (2022b) | ✗ | 62.03 | 23.13 | 53.97 | 60.35 | 35.74 | 47.04 | 43.30 |
| TPT Shu et al. (2022) | ✓ | 60.74 | 26.67 | 54.70 | 59.11 | 35.09 | 47.26 | 43.89 |
| DiffTPT Feng et al. (2023) | ✓ | 60.80 | 31.06 | 55.80 | 58.80 | 37.10 | 48.71 | 45.69 |
| DPE Zhang et al. (2024a) | ✓ | 63.41 | 30.15 | 56.72 | 63.72 | 40.03 | 50.81 | 47.66 |
| TDA Karmanov et al. (2024) | ✗ | 61.35 | 30.29 | 55.54 | 62.58 | 38.12 | 49.58 | 46.63 |
| **SOBA (Ours)** | ✗ | 62.30 | **34.09** | **57.22** | **63.81** | 39.59 | **51.22** | **48.68** |
| *(b) Full results on the OOD Benchmark with ViT-B/16 backbone* | | | | | | | | |
| CLIP | ✗ | 68.34 | 49.89 | 61.88 | 77.65 | 48.24 | 61.20 | 59.42 |
| CoOp Zhou et al. (2022b) | ✓ | 71.51 | 49.71 | 64.20 | 75.21 | 47.99 | 61.72 | 59.28 |
| CoCoOp Zhou et al. (2022a) | ✓ | 71.02 | 50.63 | 64.07 | 76.18 | 48.75 | 62.13 | 59.91 |
| Tip-Adapter Zhang et al. (2022b) | ✗ | 70.75 | 51.04 | 63.41 | 77.76 | 48.88 | 62.37 | 60.27 |
| TPT Shu et al. (2022) | ✓ | 68.98 | 54.77 | 63.45 | 77.06 | 47.94 | 62.44 | 60.81 |
| DiffTPT Feng et al. (2023) | ✓ | 70.30 | 55.68 | 65.10 | 75.00 | 46.80 | 62.28 | 60.52 |
| MTA Zanella & Ben Ayed (2024) | ✓ | 70.08 | 58.06 | 64.24 | 78.33 | 49.61 | 64.06 | 62.56 |
| VTE Döbler et al. (2024) | ✓ | 60.40 | 62.70 | 65.10 | 80.40 | 50.20 | 64.60 | 63.76 |
| DPE Zhang et al. (2024a) | ✓ | 71.91 | 59.63 | 65.44 | 80.40 | 52.26 | 65.93 | 64.43 |
| TPS Sui et al. (2025) | ✓ | 71.45 | 60.61 | 64.91 | 80.20 | 50.88 | 65.61 | 64.15 |
| TDA Karmanov et al. (2024) | ✗ | 69.51 | 60.11 | 64.67 | 80.24 | 50.54 | 65.01 | 63.89 |
| LDA* | ✗ | 69.96 | 60.06 | 64.56 | 80.09 | 49.57 | 64.85 | 63.57 |
| PCA* | ✗ | 70.32 | 61.42 | 65.22 | 80.64 | 50.81 | 65.28 | 64.52 |
| **SOBA (Ours)** | ✗ | 71.09 | **63.27** | **66.08** | **81.35** | **53.06** | **66.97** | **65.94** |

Table 1: **Results on the OOD Benchmark**. Our method performs best on both backbones. The best results are in **bold** and the second-best results are underlined. *OOD average* refers to the average accuracy on the four OOD datasets from ImageNet, while *average* refers to the average accuracy across all datasets. LDA* and PCA* refer to the variants constructed by augmenting our baseline with other transformation methods.

where $\alpha$ is a hyperparameter controlling the contribution of the SOBA transformation.

**Analysis.** The core advantage of SOBA lies in its direct and precise transformation of class prototypes, distinguishing it from general projection methods such as PCA. By reshaping the prototypes that define the decision boundary, SOBA enhances class separability. Since prototypes are statistical averages of high-confidence samples, they naturally smooth out individual noise and high-entropy perturbations Weinberger & Saul (2009). This targeted adjustment makes SOBA both more efficient and more effective than conventional methods, as further validated in Appendix C and D and subsequent experiments.

## 4 EXPERIMENT

### 4.1 EXPERIMENTAL SETUP

**Benchmarks.** Based on previous work Shu et al. (2022); Feng et al. (2023); Karmanov et al. (2024); Zanella & Ben Ayed (2024), we selected the out-of-distribution (OOD) benchmark and the cross-dataset benchmark as the foundational experiments for our study.

**1).** For the **OOD benchmark**, we test the effectiveness of our method on out-of-distribution datasets using ImageNet and its four OOD sub-datasets, which include ImageNet-A Hendrycks et al. (2021b), ImageNet-R Hendrycks et al. (2021a), ImageNet-V2 Recht et al. (2019), and ImageNet-S Wang et al. (2019). The purpose of the OOD benchmark is to evaluate the model's generalization ability to data from the same class but different domain distributions.

**2).** For the **cross-dataset benchmark**, we use 10 public datasets to evaluate the cross-dataset classification capability of our method. Each dataset comes from different classes and domains, including: Aircraft Maji et al. (2013), Caltech101 Fei-Fei et al. (2004), Car Krause et al. (2013), DTD Cimpoi et al. (2014), EuroSAT Helber et al. (2019), Flowers102 Nilsback & Zisserman (2008), Food101 Bossard et al. (2014), Pets Parkhi et al. (2012), SUN397 Xiao et al. (2010), and UCF101 Soomro et al. (2012).

**Comparison Methods.** We compare our method with zero-shot CLIP Radford et al. (2021), CoOp Zhou et al. (2022b), CoCoOp Zhou et al. (2022a), Tip-Adapter Zhang et al. (2022b), and other state-of-the-art (SOTA) methods in the TTA domain that do not require a training set, such as

| Method | Aircraft | Caltech101 | Cars | DTD | EuroSAT | Flower102 | Food101 | Pets | SUN397 | UCF101 | *Average* |
|---|---|---|---|---|---|---|---|---|---|---|---|
| *(a) Full results on the Cross-Dataset Benchmark with ResNet50 backbone* | | | | | | | | | | | |
| CLIP Radford et al. (2021) | 16.11 | 87.26 | 55.89 | 40.37 | 25.79 | 62.77 | 74.82 | 82.97 | 60.85 | 59.48 | 56.63 |
| CoOp Zhou et al. (2022b) | 15.12 | 86.53 | 55.32 | 37.29 | 26.20 | 61.55 | 75.59 | 87.00 | 58.15 | 59.05 | 56.18 |
| CoCoOp Zhou et al. (2022a) | 14.61 | 87.38 | 56.22 | 38.53 | 28.73 | 65.57 | 76.20 | 88.39 | 59.61 | 57.10 | 57.23 |
| TPT Shu et al. (2022) | 17.58 | 87.02 | 58.46 | 40.84 | 28.33 | 62.69 | 74.88 | 84.49 | 61.46 | 60.82 | 57.66 |
| DiffTPT Feng et al. (2023) | 17.60 | 86.89 | 60.71 | 40.72 | 41.04 | 63.53 | 79.21 | 83.40 | 62.72 | 62.67 | 59.85 |
| HisTPT Zhang et al. (2024b) | 18.10 | 87.20 | 61.30 | 41.30 | 42.50 | 67.60 | **81.30** | 84.90 | 63.50 | 64.10 | 61.18 |
| DPE Zhang et al. (2024a) | **19.80** | **90.83** | 59.26 | **50.18** | 41.67 | 67.60 | 77.83 | 85.97 | 64.23 | 61.98 | 61.93 |
| TDA Karmanov et al. (2024) | 17.61 | 89.70 | 57.78 | 43.74 | 42.11 | **68.74** | 77.75 | 86.18 | 62.53 | 64.18 | 61.03 |
| **SOBA (Ours)** | 19.05 | 90.39 | **62.38** | 45.62 | **43.30** | 68.11 | 79.31 | **89.05** | **66.03** | **67.90** | **63.11** |
| *(b) Full results on the Cross-Dataset Benchmark with ViT-B/16 backbone* | | | | | | | | | | | |
| CLIP Radford et al. (2021) | 23.22 | 93.55 | 66.11 | 45.04 | 50.42 | 66.99 | 82.86 | 86.92 | 65.63 | 65.16 | 64.59 |
| CoOp Zhou et al. (2022b) | 18.47 | 93.70 | 64.51 | 41.92 | 46.39 | 68.71 | 85.30 | 89.14 | 64.15 | 66.55 | 63.88 |
| CoCoOp Zhou et al. (2022a) | 22.29 | 93.79 | 64.90 | 45.45 | 39.23 | 70.85 | 83.97 | 90.46 | 66.89 | 68.44 | 64.63 |
| TPT Shu et al. (2022) | 24.78 | 94.16 | 66.87 | 47.75 | 42.44 | 68.98 | 84.67 | 87.79 | 65.50 | 68.04 | 65.10 |
| DiffTPT Feng et al. (2023) | 25.60 | 92.49 | 67.01 | 47.00 | 43.13 | 70.10 | 87.23 | 88.22 | 65.74 | 62.67 | 65.47 |
| MTA Zanella & Ben Ayed (2024) | 25.20 | 94.21 | 68.47 | 45.90 | 45.36 | 68.06 | 85.00 | 88.24 | 66.60 | 68.69 | 65.58 |
| HisTPT Zhang et al. (2024b) | 26.90 | 94.50 | 69.20 | 48.90 | 49.70 | 71.20 | **89.30** | 89.10 | 67.20 | 70.10 | 67.61 |
| VTE Döbler et al. (2024) | 24.10 | 93.30 | 69.00 | 47.30 | 47.60 | 65.50 | 83.40 | 87.00 | 66.50 | 67.00 | 65.07 |
| DPE Zhang et al. (2024a) | **28.95** | 94.81 | 67.31 | **54.20** | 55.79 | **75.07** | 86.17 | 91.14 | 70.07 | 70.44 | 69.40 |
| TPS Sui et al. (2025) | 26.34 | 95.09 | 69.06 | 50.47 | 44.48 | 71.54 | 85.23 | 87.35 | 68.98 | 71.00 | 66.96 |
| TDA Karmanov et al. (2024) | 23.91 | 94.24 | 67.28 | 47.40 | 58.00 | 71.42 | 86.14 | 88.63 | 67.62 | 70.66 | 67.53 |
| LDA* | 23.87 | 94.19 | 67.21 | 47.36 | 57.97 | 71.38 | 86.09 | 88.59 | 67.57 | 70.61 | 67.48 |
| PCA* | 24.28 | 94.45 | 67.62 | 48.10 | 58.64 | 71.77 | 86.49 | 88.97 | 68.01 | 71.05 | 67.94 |
| **SOBA (Ours)** | 28.07 | **95.02** | **71.49** | 47.24 | **61.90** | 71.93 | 87.52 | **92.86** | **71.11** | **74.28** | **70.14** |

Table 2: **Results on the Cross-Dataset Benchmark.** Our method achieves the highest average accuracy on both backbones. The best results are in **bold** and the second-best results are underlined. *Average* refers to the average accuracy across all datasets. LDA* and PCA* refer to the variants constructed by augmenting our baseline with other transformation methods.

TPT Shu et al. (2022), DiffTPT Feng et al. (2023), MTA Zanella & Ben Ayed (2024), HisTPT Zhang et al. (2024b), TDA Karmanov et al. (2024), etc.

**Implementation Details**. Our method is built upon pre-trained CLIP Radford et al. (2021), where the text encoder is a Transformer Vaswani (2017) and the image encoder can be either ResNet He et al. (2016) or ViT Dosovitskiy (2020). Since our method is training-free, all text prompts follow ZS-CLIP (e.g., "a photo of a <classname>"). We set the batch size to 1 for constructing the dynamic queue, and evaluate all experiments with top-1 accuracy on an NVIDIA Quadro RTX 6000 GPU. Our experimental results are reported

| Method | Training-free | Testing Time | OOD Average | Improved |
|---|---|---|---|---|
| CLIP-ResNet-50 | ✓ | **12min** | 43.09 | 0. |
| TPT | ✗ | 12h 50min | 43.89 | +0.80 |
| DiffTPT | ✗ | 34h 45min | 45.69 | +2.60 |
| DPE | ✗ | 1h 19min | 47.66 | +4.57 |
| TDA | ✓ | 16min | 46.63 | +3.54 |
| **SOBA (Ours)** | ✓ | 13min 46s | **48.68** | **+5.59** |

Table 3: Comparisons of our method with CLIP-ResNet-50, TPT, DiffTPT, and TDA in terms of efficiency and accuracy. The experiments are conducted on the OOD benchmark. Test-time represents the run-time on ImageNet, and all were conducted on the NVIDIA Quadro RTX 6000.

as the averages over five runs with different random seeds, with variations within **±0.3%**. More details on hyperparameter search and queue length selection are provided in Appendix F.1.

## 4.2 COMPARISON WITH STATE-OF-THE-ARTS

Like TPT, DiffTPT, MTA, and TDA, we evaluate our method on both the **OOD benchmark** and the **cross-dataset benchmark**.

**Results on the Out-of-Distribution Benchmark.** Table 1 provides a comparison between our method and state-of-the-art (SOTA) approaches across different backbones on ImageNet and four out-of-distribution (OOD) datasets. Our method surpasses existing approaches on all OOD datasets. Notably, it outperforms TDA with an increase of **2.05%** in OOD average accuracy using the ResNet-50 backbone and **2.05%** with the ViT-B/16 backbone. Additionally, our approach demonstrates a significant **1.79%** improvement over TPS with the ViT-B/16 backbone. These results affirm

| Method | ImageNet-A | ImageNet-V2 | ImageNet-R | ImageNet-S | *OOD Average* | test-time on ImageNet |
|---|---|---|---|---|---|---|
| Baseline | 29.04 | 53.20 | 61.16 | 37.39 | 45.20 | - |
| +SOBA w/o GDA | 33.70 | **57.37** | 62.47 | 39.26 | 48.20 | 28min 09s |
| +SOBA | **34.09** | 57.22 | **63.81** | **39.59** | **48.68** | 13min 46s |

Table 4: Performance improvement of our method over cache baseline on OOD benchmark. " SOBA w/o GDA" denotes our proposed method SOBA without employing the GDA assumption.

the effectiveness of exploring new decision boundaries beyond the original CLIP decision surface, validating our approach.

**Efficiency Comparison.** As shown in Table 3, we evaluate the efficiency of our method using ResNet-50 as the backbone and compare it with several existing test-time adaptation methods on ImageNet, focusing on inference speed and accuracy. The performance metrics for CLIP-ResNet-50, TPT, DiffTPT, DPE, and TDA are taken from the TDA paper. While our method incurs a minor efficiency cost compared to zero-shot CLIP, it achieves a **2.49% accuracy improvement**. Unlike TPT and DiffTPT, which require backpropagation, our approach is substantially faster—approximately **56× faster than TPT** and **151× faster than DiffTPT**—while still delivering competitive accuracy. Compared to DPE, our method is roughly **5.7× faster** (13min 46s vs. 1h 19min) with only a small 1.11% accuracy drop, demonstrating a favorable speed-accuracy trade-off. Relative to TDA, our approach improves both efficiency and accuracy, reducing inference time by 2min 14s while increasing accuracy by 0.95%. These results highlight that our method provides a highly efficient and practical solution for test-time adaptation, achieving a strong balance between speed and performance.

**Results on the Cross-Datasets Benchmark.** To further validate the feasibility and effectiveness of our method, we compared it with state-of-the-art (SOTA) methods on 10 datasets from different classes and domains. As shown in Table 2, our method achieves superior performance across two backbones. When using ResNet-50 as the backbone, our method attains Top-2 results on all 10 datasets and improves the average accuracy by 2.08% compared to TDA. With ViT-B/16 as the backbone, our method achieves Top-2 results on 9 out of 10 datasets and increases the average accuracy by **2.61% relative to TDA**. The performance on the cross-dataset benchmark further demonstrates that our method remains effective even when faced with datasets from different classes and domains. Moreover, our method does not require additional training or backpropagation on both benchmarks, making it well-suited for testing adaptation tasks with CLIP.

## 4.3 ANALYSIS

**Comparison with Other Transformation Methods.** To evaluate the efficacy of SOBA's basis transformation, we compare it with classical linear methods, LDA and PCA, as shown in Table 2 and 1. SOBA consistently outperforms both across all OOD benchmarks, with OOD average improvements over CLIP of +4.11% (baseline), +5.43% (LDA), +5.86% (PCA), and **+6.52%** (SOBA), highlighting its superior effectiveness.

**1). Comparison with LDA.** SOBA employs SVD to perform a label-independent basis transformation, projecting data onto orthogonal principal directions that **decouple feature information**. This enables robust domain adaptation even in high-entropy environments. In contrast, LDA is a supervised method that **relies highly on class labels** to compute between- and within-class scatter matrices. Noisy pseudo-labels can **distort these matrices**, causing the projection to deviate from the true discriminative subspace and consequently degrading performance in test-time adaptation. For further explanations and additional experiments, please refer to the Appendix C.1.1.

**2). Comparison with PCA.** PCA, on the other hand, is unsupervised and focuses on global variance, without explicitly enhancing class separability. SOBA combines the flexibility of unsupervised transformation with a task-adaptive mechanism to refine downstream classification. Notably, even on high-entropy datasets such as ImageNet-S, SOBA achieves a 4.82% gain over CLIP, demonstrating robustness even when pseudo-labels are unreliable.

**Comparison with Other Classifiers.** In Fig. 3(a), we present a comparison of our method with other classifiers. Due to changes in the feature space, directly minimizing the Manhattan (L1) distance and Euclidean (L2) distance to class centers is no longer applicable, and it even results in degradation compared to zero-shot CLIP. Our method, compared to the basic NCM classifier, achieves better decision boundaries by utilizing the reoriented space, further addressing the TTA problem.

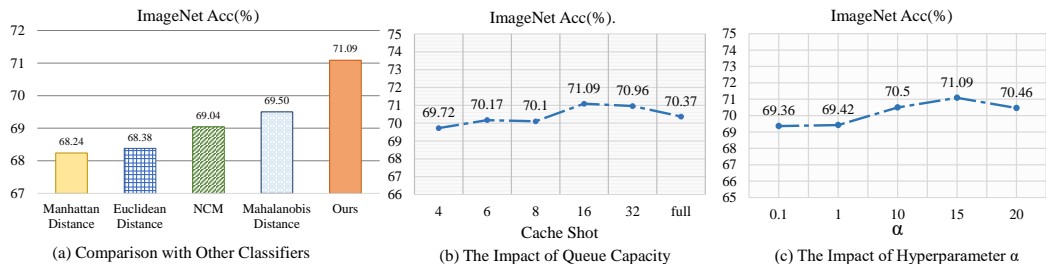

(a) Comparison with Other Classifiers  (b) The Impact of Queue Capacity  (c) The Impact of Hyperparameter α

Figure 3: Subfigure (a) shows a comparison with other classifiers, where our SOBA achieves the best performance. Subfigure (b) presents a study on different dynamic queue lengths. Subfigure (c) presents a study on the impact of the hyperparameter $\alpha$.

**Feasibility of the GDA Assumption.** To assess the feasibility of the GDA assumption in the TTA setting, we conducted an additional experiment in which each pseudo-labeled class was assigned its own covariance matrix (i.e., the downstream distribution does not satisfy the GDA assumption). The results are summarized in Table 4. We observe that the choice of whether or not to adopt the GDA assumption has little impact on the overall inference performance. However, employing the GDA assumption—namely, sharing a common covariance matrix across all classes—significantly reduces inference time (**13m46s vs. 28m09s**). Therefore, we conclude that the GDA assumption does not compromise the generality of SOBA in the tasks considered in this work.

## 4.4 Ablation Studies

**Effectiveness of SOBA.** To clearly illustrate the effectiveness of our method, we compare it with a simple yet effective baseline. In Table 3, we report the ablation experiments on the OOD benchmark and cross-dataset benchmark, respectively. Since the baseline method also does not involve backpropagation and is based on the original CLIP feature space, comparing it with this baseline allows us to directly observe the pure benefit of the space reorienting provided by SOBA.

As shown in Table 4, compared to the baseline, our method achieves substantial improvements across almost all datasets. Specifically, on the OOD benchmark, our evaluation metric *OOD average* improves by **2.41%** over the baseline. When combined with the comparisons to TDA in Section 4.2, which rely on the original CLIP feature space, these results indicate that applying a basis transformation to reconstruct the original feature space is a feasible and effective solution for addressing the TTA problem, yielding superior performance to using the original CLIP features. Detailed results are provided in the Appendix C.2.

**Hyperparameter Sensitivity Analysis. 1). Queue Capacity K.** Fig. 3(b) shows the effect of dynamic queue capacity. Accuracy first increases and then decreases as capacity grows: a small queue stores highly representative features, while a larger queue may include easily confusable features, affecting subsequent predictions. We set 16 as the per-class storage limit on the OOD benchmark. Ablation results for the Cross-Dataset benchmark are provided in Appendix C.3. **2). Hyperparameter $\alpha$.** In Fig. 3(c), we illustrate the impact of $\alpha$ from 13. Based on the performance on ImageNet, we ultimately select $\alpha = 15$ as the final value. For the effect of $\alpha$ on other datasets, please refer to the Appendix C.3.

## 5 Conclusion

In this work, we introduce a space reorienting with basis transformation (SOBA) method, which re-expresses class prototypes in a test-induced orthogonal basis, effectively enlarging inter-class margins and sharpening decision boundaries to improve recognition of semantically similar categories without modifying features, prompts, or model parameters. Experimental results across various benchmarks have demonstrated that our method not only outperforms state-of-the-art approaches but is also easy to implement and highly efficient. Detection and semantic segmentation tasks can still be regarded as fine-grained classification tasks. **In future work**, we plan to extend the application of SOBA to related domains to validate its effectiveness across various visual tasks.

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

## A  Limitations

Detection and semantic segmentation tasks can still be regarded as fine-grained classification tasks. In future work, we plan to extend the application of SOBA to related domains to validate its effectiveness across various visual tasks.

## B  The Use of Large Language Models

In this work, a large language model (LLM) was solely employed for linguistic refinement of the manuscript. Specifically, the LLM assisted in polishing phrasing, grammar, and readability of sentences. No substantive content, technical analyses, or experimental results were generated or altered by the LLM; all scientific ideas, methodology, and data interpretation were independently conceived and conducted by the authors.

## C  Additional Ablation Study and Analysis

### C.1  Additional Robustness Analysis

We elucidate the robustness of SOBA from three complementary perspectives. First, we analyze its **behavior under high-entropy conditions** C.1.1, where feature representations are inherently ambiguous. Second, we examine its stability in **the presence of noisy pseudo-labels** C.1.2, ensuring that the transformation remains reliable even when supervision is imperfect. Third, we discuss the **feasibility of the Gaussian Discriminant Analysis (GDA) assumption** C.1.3, which underpins the efficiency and effectiveness of our framework. Finally, we analyze the **dependence on initial pseudo-labels** in C.1.4 and demonstrate that our method is inherently designed to handle pseudo-label noise, with both its mechanism and empirical evaluation showing strong robustness.

### C.1.1  Discussion in High-entropy Environments

**Our SOBA:** On inherently challenging datasets with high pseudo-label entropy (e.g., Aircraft), SOBA still achieves significant performance gains over other methods. This indirectly demonstrates that even under high initial prediction uncertainty, our approach can effectively uncover and exploit the separable structure within the data.

| Method | UCF101 | Cars | Aircraft | Average |
|--------|--------|-------|----------|---------|
| CLIP   | 65.16  | 66.11 | 23.22    | 64.59   |
| TDA    | 70.66  | 67.28 | 23.91    | 67.53   |
| LDA$^*$ | 70.61 | 67.21 | 23.87    | 67.48   |
| SOBA   | **74.28** | **71.49** | **28.07** | **70.14** |

Table 5: **Performance on High-Entropy Datasets (hard Classes).** "Average" represents the mean performance on the Cross-Dataset benchmark.

**Comparison with TDA.** For example, as shown in Table 5, on the highly entropic Aircraft dataset, SOBA still achieves improvements similar to the overall average over TDA, demonstrating its robustness.

**Comparison with LDA.** SOBA leverages SVD to perform a basis transformation, effectively changing the coordinate system and mapping the data from one basis (e.g., the standard orthogonal basis) to a selected orthogonal basis. In this process, SVD generates mutually orthogonal principal directions, which decouple the original information. As a result, even in high-entropy settings, SOBA can robustly enhance domain adaptation capabilities.

In contrast, LDA is a supervised discriminative transformation designed to find a projection space where samples from different classes are maximally separated while samples within the same class remain tightly clustered. This transformation relies on class labels to compute the between-class and within-class scatter matrices, which determine the projection directions.

Due to this dependency, LDA is highly sensitive to the quality of pseudo-labels. In unsupervised or weakly supervised scenarios, noisy pseudo-labels can lead to underestimation of between-class scatter and overestimation of within-class scatter, causing the LDA projection directions to deviate from the true optimal discriminative subspace and thereby degrading performance in test-time adaptation tasks.

### C.1.2 STABILITY AND IMPACT OF NOISE

To verify the robustness of SOBA against noisy pseudo-labels, we injected Gaussian white noise into the pseudo-labels during testing (Table 6). Under low noise ratios (20% and 40%), SOBA exhibits a strong ability to correct the decision boundaries. Even when the noise ratio exceeds 50%, SOBA still surpasses the original CLIP, indicating its resilience in constructing clear decision boundaries under noisy conditions. This robustness can be attributed to the fact that the principal directions extracted via SVD capture the dominant intrinsic structure and variation trends of

| Noise Ratio | Accuracy | Improved over CLIP |
|---|---|---|
| SOBA w 60% | 61.68 | 1.87 |
| SOBA w 40% | 61.95 | 2.14 |
| SOBA w 20% | 62.12 | 2.31 |
| SOBA | **62.30** | **2.49** |

Table 6: **Analysis of different pseudo-label noise ratios.** The experiments are conducted on ImageNet.

the data. Since the transformation is derived from the global distribution, it possesses a natural immunity to small amounts of incorrect pseudo-labels, as individual noisy samples are unlikely to alter the overall statistical structure. This sharply contrasts with methods such as LDA, which are highly sensitive to every label.

### C.1.3 FEASIBILITY OF THE SHARED GAUSSIAN ASSUMPTION

GDA (Gaussian Discriminant Analysis) assumes that all class distributions share the same covariance matrix. This assumption is made to simplify the covariance update computations, thereby reducing inference time during testing.

Table 4 shows SOBA's strong performance on datasets with **significant domain shifts**, demonstrating that the GDA assumption is valid even under severe domain shifts. Additionally, we conducted experiments where **each pseudo-label class has its own covariance matrix** (i.e., the downstream distributions do not satisfy the GDA assumption). The results are summarized in the Table 4. We observe that adopting the GDA assumption has little impact on overall test-time performance. However, using the GDA assumption, which shares the same covariance matrix across all classes, significantly reduces inference time (**13 min 46 s vs. 28 min 09 s**). Therefore, we conclude that the GDA assumption does not compromise the generality of SOBA for the tasks considered in this work.

### C.1.4 ANALYSIS OF DEPENDENCE ON INITIAL PSEUDO-LABELS

Our method was designed with the issue of pseudo-label noise in mind, and both its mechanism and empirical evaluations demonstrate strong robustness.

**Mechanism-level robustness.** The basis transformation in SOBA does not depend on any single pseudo-labeled sample. Instead, it is derived from the *global statistical properties* of all features stored in the dynamic queue, i.e., the shared covariance structure of the data. The "principal directions" extracted via SVD capture the dominant structure and variation trends of the feature distribution. Such global statistics are naturally resilient to a small fraction of incorrect pseudo-labels, as individual noisy samples have little influence on the overall distribution. This stands in sharp contrast to methods such as LDA, which are highly sensitive to individual label assignments.

**Empirical validation. Direct noise injection.** In the appendix, we conduct rigorous robustness tests by artificially injecting different levels of Gaussian white noise into pseudo-labels (ranging from 20% to 60%). Results show that **even under 60% noise, SOBA still significantly outperforms the original CLIP model**, directly demonstrating its effectiveness in noisy label environments.

**Performance on high-entropy datasets.** On particularly challenging datasets with inherently high pseudo-label entropy, such as *Aircraft*, SOBA consistently achieves substantial improvements over competing methods. This indirectly confirms that, even under high prediction uncertainty, SOBA is capable of uncovering and leveraging discriminative structure in the data.

**Conclusion.** Although SOBA leverages pseudo-labels to construct the dynamic queue, **its core geometric transformation relies on robust estimates of the global distribution**. Consequently, SOBA exhibits strong resilience to pseudo-label noise and remains reliable under uncertain test-time adaptation scenarios.

(a) Performance improvement of our method over cache baseline on OOD benchmark.

| Method | ImageNet | ImageNet-A | ImageNet-V2 | ImageNet-R | ImageNet-S | *Average* | *OOD Average* |
|---|---|---|---|---|---|---|---|
| Baseline | 69.04 | 60.04 | 64.54 | 80.16 | 49.39 | 64.63 | 63.53 |
| **+SOBA (Ours)** | **71.09** | **63.27** | **66.08** | **81.35** | **53.06** | **66.97** | **65.94** |
| Improvement | **+2.05** | **+3.23** | **+1.54** | **+1.19** | **+3.67** | **+2.38** | **+2.41** |

(b) Performance improvement of our method over cache baseline on Cross-Dataset benchmark.

| Method | Aircraft | Caltech101 | Cars | DTD | EuroSAT | Flower102 | Food101 | Pets | SUN397 | UCF101 | *Average* |
|---|---|---|---|---|---|---|---|---|---|---|---|
| Baseline | 24.72 | 94.07 | 67.79 | 45.80 | 55.06 | 71.15 | 86.4 | 88.41 | 67.69 | 70.24 | 67.13 |
| **+SOBA (Ours)** | **28.07** | **95.02** | **71.49** | **47.24** | **61.90** | **71.93** | **87.52** | **92.86** | **71.11** | **74.28** | **70.14** |
| Improvement | **+3.35** | **+0.95** | **+3.70** | **+1.44** | **+6.84** | **+0.78** | **+1.12** | **+4.45** | **+3.42** | **+4.04** | **+3.01** |

Table 7: **Performance improvement of our method over cache baseline on both benchmarks.** The experiments employ ViT-B/16 as the backbone.

## C.2 ADDITIONAL EXPERIMENTS TO VALIDATE THE METHOD'S EFFECTIVENESS

As shown in Table 7, compared to baseline, our work demonstrates significant improvements across nearly all datasets in both benchmarks. Compared to the baseline, on the OOD benchmark, our two evaluation metrics, *average* and *OOD average*, improved by **1.6%** and **1.53%**. On the cross-dataset benchmark, we achieved a **2.19%** improvement in *average*. Combining our finding with the comparisons to TDA in Section 4.2, which rely on the original CLIP feature space, we can conclude that applying a basis transformation to change the original space is a feasible solution to address the TTA problem, and it achieves better performance than the original CLIP feature space.

## C.3 ADDITIONAL HYPERPARAMETER AENSITIVITY ANALYSIS

| K | Aircraft | Caltech101 | Cars | DTD | EuroSAT | Flower102 | Food101 | Pets | SUN397 | UCF101 | *Average* |
|---|---|---|---|---|---|---|---|---|---|---|---|
| 2 | 27.17 | 94.01 | 68.16 | 46.17 | 57.52 | 71.61 | 87.23 | 91.99 | 68.27 | 73.25 | 68.52 |
| 4 | 27.44 | 94.33 | 69.27 | 45.70 | 56.76 | 71.77 | 87.42 | 92.01 | 68.63 | 72.56 | 68.57 |
| 6 | 27.53 | 94.66 | 70.63 | 45.76 | 57.09 | 71.81 | 87.52 | 91.66 | 69.78 | 73.09 | 68.93 |
| 8 | 27.77 | **95.13** | 70.54 | 46.35 | 61.25 | **71.95** | 87.40 | 92.15 | 69.89 | 73.99 | 69.61 |
| 16 | **28.07** | 95.02 | 71.49 | **47.24** | **61.90** | 71.93 | 87.52 | **92.86** | **71.11** | **74.28** | **70.14** |
| 32 | 27.72 | 93.73 | **71.59** | 46.71 | 60.74 | 71.65 | **87.62** | 93.18 | 69.83 | 72.93 | 69.55 |
| full | 27.66 | 93.73 | 71.01 | 46.18 | 60.57 | 71.65 | 87.36 | 93.12 | 69.39 | 72.71 | 69.32 |

Table 8: **Results on the Cross-Dataset Benchmark.** The performance of SOBA with different K on the Cross-Dataset benchmark. Due to the complexity of the datasets in the cross-dataset benchmark, the performance of each dataset may vary differently as the queue capacity increases. The backbone used in the experiments is ViT-B.

This section supplements the ablation experiment on queue capacity and hyperparameter $\alpha$.

**Queue Capacity K.** For the Pets dataset Parkhi et al. (2012), the best accuracy is achieved when the queue capacity per class is 32. We believe the reason is that the differences between different classes in the Pets dataset are significant, as these classes not only exhibit distinct visual features (such as fur color, shape, and body size), but also show considerable diversity in terms of image background, posture, and camera angle. Therefore, increasing the queue capacity can better capture the information of the feature space, allowing the reconstructed basis and class prototypes to more effectively reflect the differences between classes. Finally, we used $K = 16$ as the overall queue capacity for the cross-dataset benchmark.

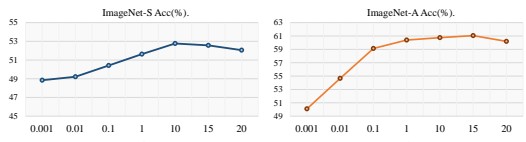

Figure 4: The Impact of Hyperparameter $\alpha$.

**Hyperparameter $\alpha$.** For the hyperparameter $\alpha$, as illustrated in Fig. 4, we conducted a hyperparameter search on ImageNet to identify an effective value and ultimately **set $\alpha = 15$ for all subsequent benchmarks**.

**Discussion of fixed hyperparameters across different datasets.** The primary justification for fixing hyperparameters (e.g., $\alpha$ and $k$) in our method is to ensure **experimental consistency** and **comparability**. (1) Ensuring Consistency and Comparability: By setting the same hyperparameters across all experiments, we ensure that results on different datasets are obtained under identical, controlled conditions. This allows for direct comparison and enables us to draw reliable conclusions about the general effectiveness of our method. (2) Avoiding Excessive Tuning: Searching for the optimal hyperparameter values for each dataset would introduce an additional and potentially open-ended optimization process. Fixing these parameters simplifies the experimental procedure, making it more practical and reproducible. (3) Reflecting Intrinsic Effectiveness: This decision is intended to demonstrate the intrinsic performance of our method, rather than showing that it can achieve better results through dataset-specific optimization. It proves that the method itself is robust and not dependent on overfitting to individual datasets.

**Basis Vector Update Frequency** We conducted experiments on the Aircraft dataset using two update strategies: updating the covariance matrix for every batch, updating every 20% of the dataset, and our adopted strategy of updating every 10%. The results are summarized in the Table 9.

We observe that the dynamically updated version of SOBA (per batch) does not outperform SOBA updated every 10% of the dataset. This is because the randomness of test samples in the TTA setting introduces stochasticity into covariance updates, resulting in large fluctuations in accuracy during the early stages.

| Update Strategy | Aircraft |
|---|---|
| SOBA (batch) | 27.89 |
| SOBA (20%) | 27.70 |
| SOBA (10%) | 28.07 |

Table 9: Performance of SOBA under different basis update frequencies on the Aircraft dataset.

As for SOBA updated every 20% of the dataset (SOBA-20%), its lower accuracy can be attributed to the reliance on the original CLIP predictions for the first 20% of samples. Furthermore, the longer update interval hinders timely incorporation of the updated queue for basis vector construction, thereby impeding effective reconstruction of the feature space.

# D  SOBA IS NEITHER LDA NOR PCA

SOBA is fundamentally different from both LDA and PCA: although it is inspired by linear transformations, its objective, design, and application are uniquely tailored to test-time adaptation.

## D.1  WHY SOBA IS NOT SIMPLE LDA?

SOBA is **not a special case of LDA**; the two methods differ fundamentally in both concept and implementation.

**1). Objective function**: As noted in Section 4.3, LDA is a supervised learning method that seeks a projection maximizing inter-class scatter while minimizing intra-class scatter. In contrast, SOBA is based on PCA and follows an unsupervised philosophy, aiming to preserve as much variance in the data as possible without directly using class labels to optimize class separability.

**2). Information dependency**: LDA requires the computation of intra-class and inter-class scatter matrices, which heavily depend on class labels. SOBA, on the other hand, only computes a covariance matrix shared across all classes.

**3). Experimental comparison**: The paper compares SOBA and LDA as two distinct methods (see the LDA* rows in Tables 1 and 2). Results show that SOBA outperforms LDA variants across all benchmarks, demonstrating that they are indeed different methods and that SOBA is more effective for this task.

**4). Effectiveness of SOBA**: Its efficacy arises from a PCA-inspired orthogonal basis transformation that changes the feature space to better align with the intrinsic structure of the data, thereby defining clearer decision boundaries in the new space. It is not a special case of LDA, and because the

transformation relies more on the global data distribution than on precise label information, SOBA exhibits greater robustness to pseudo-label noise during testing.

### D.2 Why SOBA Is Not Simple PCA?

The innovation of SOBA **does not lie in the mere use of SVD**, but in its highly strategic application. Unlike applying a generic PCA projection to all features, we selectively transform only the class prototypes. This design is superior for two main reasons:

First, it enables **precise reshaping of decision boundaries**. Since the geometry of the classifier is determined by the class prototypes, directly transforming them provides the most efficient and direct path to enhancing class separability Mensink et al. (2013a). Second, as prototypes are computed as the averages of multiple samples, **they inherently possess statistical stability**, making our transformation more robust to noisy individual samples. This idea of learning an optimal feature space for classification aligns conceptually with the core objective of metric learning Weinberger & Saul (2009). Therefore, our approach is more principled and effective than a generic feature projection.

In summary, SOBA is not a mere application of PCA, but rather a PCA-inspired geometric transformation framework tailored for TTA tasks. By reshaping the representation of class prototypes in a new coordinate system, it precisely addresses the challenge of blurred decision boundaries caused by fixed feature spaces in TTA.

## E  Additional Implementation of SOBA

In this section, we provide a detailed description of the overall process of handling the feature space with basis vectors in our SOBA method.

### E.1 SOBA Process

The SOBA process includes the following key steps: for each test sample $x_{test}$, the algorithm first extracts the image feature $f_{test}$ and text features $W_t$ using CLIP's visual encoder $g_v(\theta_v)$ and text encoder $g_t(\theta_t)$, and calculates the original CLIP logits by Eq. 1. It then generates pseudo-labels by applying one-hot encoding to the original logits by Eq. 2, and updates the dynamic queue, which stores the image features, pseudo-labels, and logits. After that, we compute the prototype for each pseudo-class and calculate the covariance matrix of the queue by Eq. 10 and Eq. 11.

Next, the prototypes are reconstructed using the SOBA method to obtain new class prototypes by Eq. 12, and the transformed logits are computed based on these changed prototypes by Eq. 13. Finally, the algorithm combines the original logits and the transformed logits with a weighting factor $\alpha$ to produce the final prediction. It is worth noting that to ensure the stability and accuracy of the obtained orthogonal basis and class prototypes, we update the prototypes every 10% of the test samples. This strategy allows the algorithm to optimize the model's adaptability while maintaining computational efficiency, and reduces the impact of bases constructed from too few samples on the final results. The overall process is presented in Algorithm 1.

### E.2 Queue Update Process

In this section, we explain how to perform enqueue and dequeue operations on the queue.

First, for each test feature $x_{test}$, the algorithm checks whether the queue $L_{l_p}^{t-1}$ corresponding to the current pseudo-label $l_p$ is full. If the queue is not full, the current feature $f_{test}$ and its corresponding pseudo-label $l_p$ are simply enqueued, generating a new queue $L^t$. If the queue is full, the algorithm first calculates the maximum entropy $H_{max}$ in the queue, which represents the average uncertainty of the current features. Then, the algorithm compares the entropy of the current feature's logits $H(logits_{ori})$ with the maximum entropy $H_{max}$. If the current feature's entropy is smaller than the maximum entropy, it indicates that the feature is more certain, and the algorithm removes the feature with the highest entropy from the queue and enqueues the current feature; otherwise, the queue remains unchanged. Finally, the algorithm returns the updated queue $L^t$, which helps manage the

updates of features and pseudo-labels, ensuring that the queue adapts to new data over time. The overall process is presented in Algorithm 2.

### E.3 AN IN-DEPTH EXPLANATION OF THE BASIS TRANSFORMATION IN SOBA

We provide a rigorous derivation of the basis transformation from the standard expansion in Eq. (4) to the analysis and synthesis forms in Eqs. (6)–(8).

**Step 1. Standard expansion.** For any $\mathbf{V} \in \mathbb{R}^{n \times d}$, with the standard basis $\mathcal{E} = \{e_{ij}\}_{1 \leq i \leq n, 1 \leq j \leq d}$, we have

$$\mathbf{V} = \sum_{i=1}^{n} \sum_{j=1}^{d} v_{ij} \, \mathbf{e}_{ij}, \tag{15}$$

where $v_{ij} = \langle \mathbf{V}, \mathbf{e}_{ij} \rangle$ is the $(i,j)$-th entry of $\mathbf{V}$. This corresponds to Eq. 4.

**Step 2. Construction of a new basis.** Let $\mathbf{P} = [p_1, \ldots, p_n] \in \mathbb{R}^{n \times n}$ and $\mathbf{Q} = [q_1, \ldots, q_d] \in \mathbb{R}^{d \times d}$ be orthogonal matrices. Define the new basis

$$\mathbf{b}_{ij} := p_i q_j^\top, \quad 1 \leq i \leq n, \, 1 \leq j \leq d. \tag{16}$$

**Step 3. Orthonormality of the new basis.** For any $b_{ij}, b_{k\ell}$,

$$\langle \mathbf{b}_{ij}, \mathbf{b}_{k\ell} \rangle = \mathrm{tr}\big((p_i q_j^\top)^\top (p_k q_\ell^\top)\big) \tag{17}$$

$$= \mathrm{tr}\big(q_j p_i^\top p_k q_\ell^\top\big) \tag{18}$$

$$= (p_i^\top p_k)\,(q_\ell^\top q_j) \tag{19}$$

$$= \delta_{ik}\,\delta_{j\ell}. \tag{20}$$

Thus $\mathcal{B} = \{\mathbf{b}_{ij}\}$ forms an orthonormal basis.

**Step 4. Expansion in the new basis.** Since $\mathcal{B}$ is an orthonormal basis,

$$\mathbf{V} = \sum_{i=1}^{n} \sum_{j=1}^{d} \langle \mathbf{V}, \mathbf{b}_{ij} \rangle \, \mathbf{b}_{ij}. \tag{21}$$

This corresponds to Eq. (5).

**Step 5. Coordinates in the new basis.** Each coefficient is given by

$$\hat{v}_{ij} = \langle \mathbf{V}, \mathbf{b}_{ij} \rangle \tag{22}$$

$$= \mathrm{tr}(\mathbf{V}^\top p_i q_j^\top) \tag{23}$$

$$= q_j^\top \mathbf{V}^\top p_i \tag{24}$$

$$= p_i^\top \mathbf{V} q_j. \tag{25}$$

Thus

$$\hat{v}_{ij} = p_i^\top \mathbf{V} q_j. \tag{26}$$

This is Eq. (7).

**Step 6. Matrix form of the analysis operator.** Collecting all coefficients into the matrix $\widehat{V} = [\hat{v}_{ij}] \in \mathbb{R}^{n \times d}$, we observe

$$\hat{\mathbf{V}} = \mathbf{P}^\top \mathbf{V} \mathbf{Q}, \tag{27}$$

since $(\mathbf{P}^\top \mathbf{V} \mathbf{Q})_{ij} = p_i^\top \mathbf{V} q_j$. This corresponds to Eq. (6).

**Step 7. Synthesis (reconstruction).** Conversely, from Eq. 21,

$$\mathbf{V} = \sum_{i=1}^{n} \sum_{j=1}^{d} \hat{v}_{ij} \, p_i q_j^\top. \tag{28}$$

This can be written in matrix form as

$$\mathbf{V} = \mathbf{P} \hat{\mathbf{V}} \mathbf{Q}^\top, \tag{29}$$

which corresponds to Eq. (8).

**Step 8. Verification.**   Substituting Eq. 27 into Eq. 29, we obtain

$$\mathbf{P}(\mathbf{P}^\top \mathbf{V}\mathbf{Q})\mathbf{Q}^\top = (\mathbf{P}\mathbf{P}^\top)\mathbf{V}(\mathbf{Q}\mathbf{Q}^\top) = \mathbf{V},$$

since $\mathbf{P}\mathbf{P}^\top = \mathbf{I}_n$ and $\mathbf{Q}\mathbf{Q}^\top = \mathbf{I}_d$. Thus, the analysis and synthesis operators are exact inverses.

## F   ADDITIONAL EXPERIMENTAL DETAILS

### F.1   ADDITIONAL IMPLEMENTATION DETAILS

Our method is built upon pre-trained CLIP Radford et al. (2021), where the text encoder is a Transformer Vaswani (2017), and the image encoder can be either ResNet He et al. (2016) or Vision Transformer (ViT) Dosovitskiy (2020). Since our method is training-free, all text prompts are manually crafted in the same manner as ZS-CLIP, such as *"a photo of a <classname>"*.

**Dynamic Queue.**   To construct the dynamic cache queue, we set the batch size to 1. The queue is updated online during testing with representative samples chosen according to the minimum entropy of CLIP predictions. For the cross-dataset benchmark, we conducted experiments with different queue lengths ($K \in \{2, 4, 6, 8, 16, 32, \text{full}\}$), and finally fixed the queue length to 16 for all reported results. Detailed comparisons of queue lengths can be found in Appendix C.

**Hyperparameter Search.**   For the OOD benchmark, we perform a grid search on ImageNet to determine hyperparameters (e.g., update frequency, entropy threshold), and directly transfer the resulting configurations to the other four OOD datasets (ImageNet-A, -V2, -R, -S) to ensure consistency and fairness.

**Evaluation Protocol.**   We report top-1 accuracy for all datasets. Our experiments are conducted on an NVIDIA Quadro RTX 6000 GPU. To ensure reproducibility, we keep all random seeds fixed, and no additional training or fine-tuning is applied to CLIP parameters.

**Additional Results.**   Comprehensive ablation studies and extended results, including the effect of hyperparameters and queue length, are provided in Appendix C.

### F.2   ADDITIONAL BENCHMARK DETAILS

In this section, we provide detailed information on the two benchmarks used in our work.

**OOD Benchmark.** OOD benchmark is used to validate the model's ability to generalize to data of the same class but with different styles, assessing its robustness and effectiveness against distributional shifts. For the OOD benchmark, we used ImageNet Deng et al. (2009) along with four OOD sub-datasets to evaluate our method's performance on out-of-distribution data. These four datasets include ImageNet-A Hendrycks et al. (2021b), ImageNet-R Hendrycks et al. (2021a), ImageNet-V2 Recht et al. (2019), and ImageNet-S Wang et al. (2019). Below, we provide a brief overview of each OOD dataset.

**1). ImageNet-A** Hendrycks et al. (2021b): ImageNet-A is a curated dataset containing 200 challenging classes of images for standard ImageNet-trained models. The dataset is composed of images from the real world that are likely to cause model misclassification, specifically selected to highlight the limitations of traditional models when recognizing out-of-distribution or adversarial samples.

**2). ImageNet-R** Hendrycks et al. (2021a): ImageNet-R is a dataset derived from ImageNet, specifically designed to test model robustness under significant changes in visual style, covering 200 classes. "R" stands for "Renditions," and the dataset includes images in a variety of artistic styles, such as paintings, cartoons, and sculptures. These images differ significantly from standard ImageNet photographs, making them particularly suitable for evaluating a model's ability to generalize beyond typical photographic representations.

**3). ImageNet-V2** Recht et al. (2019): ImageNet-V2 is a dataset designed to evaluate the consistency and robustness of models trained on the original ImageNet dataset, consisting of 1000 classes. It was created by resampling the original ImageNet categories using methods that are similar but not

identical to the original collection process. ImageNet-V2 aims to measure the generalization ability of models, as it mimics the distribution of the original dataset while incorporating new, previously unseen samples.

**4). ImageNet-S** Wang et al. (2019): ImageNet-S is a dataset derived from ImageNet, containing 1000 classes, specifically designed to evaluate a model's sensitivity to background changes and its ability to focus on salient features. "S" stands for "Sketches," and the dataset consists of black-and-white sketches of the original ImageNet classes. The simplified and abstract nature of the sketches challenges models to classify images based solely on basic contours and shapes, rather than relying on background context or texture information.

**Cross-Dataset Benchmark.** The cross-dataset benchmark consists of 10 image classification datasets, each representing a distinct domain and category, designed to evaluate the model's effectiveness and generalization capability across diverse scenarios. The benchmark includes the following datasets: Caltech101 for general image classification; OxfordPets (Pets), Stanford-Cars (Cars), Flowers102, Food101, and FGVCAircraft (Aircraft) for fine-grained image classification; EuroSAT for satellite imagery classification; UCF101 for action recognition; DTD for texture classification; and SUN397 for scene classification.

For the number of classes and the number of test samples for each dataset in both benchmarks, please refer to the table

| Dataset | Classes | Test Samples |
|---|---|---|
| OOD benchmark | | |
| ImageNet | 1,000 | 50,000 |
| ImageNet-V2 | 1,000 | 10,000 |
| ImageNet-S | 1,000 | 50,000 |
| ImageNet-A | 200 | 7,500 |
| ImageNet-R | 200 | 30,000 |
| Cross-Dataset benchmark | | |
| Aircraft | 100 | 3,333 |
| Caltech101 | 101 | 2,465 |
| Cars | 196 | 8,041 |
| DTD | 47 | 1,692 |
| EuroSAT | 10 | 8,100 |
| Flowers102 | 102 | 2,463 |
| Food101 | 101 | 30,300 |
| Pets | 37 | 3,669 |
| SUN397 | 397 | 19,850 |
| UCF101 | 101 | 3,783 |

Table 10: Datasets information.

---

**Algorithm 1** The testing loop of proposed **SOBA** method for test-time adaptation

---

1: **Input:** CLIP visual encoder $g_v(\theta_v)$, text encoder $g_t(\theta_t)$, testing dataset $D_{test}$, number of classes $N$, $N$ text descriptions $T$ of $N$ classes, original basis $\mathcal{E}$, dynamic queue $L$, hyper-parameter $\alpha$, queue capacity K.

2: **for** each test sample $x_{test}$ in $D_{test}$ **do**

3:     Image embedding: $f_{test} \leftarrow g_v(\theta_v, x_{test})$

4:     Text embeddings: $W_t \leftarrow g_t(\theta_t, T)$

5:     CLIP logits: $logits_{clip} \leftarrow f_{test}W_t^{\mathrm{T}}$

6:     Pseudo-label of $x_{test}$: $l_p \leftarrow \mathtt{OneHot}(logits_{clip})$

7:     $L \leftarrow \mathtt{Update}(L, f_{test}, l_p, logits_{clip})$                 ▷ See Algorithm 2

8:     **for** each pseudo-class $l_{pk}$ in $L$ **do**

9:         Get prototype of class $l_{pk}$: $\mu_k \leftarrow \frac{\sum_{i=1}^{M_k} \mathbb{I}_{l_p=k} f_{test,i}}{\sum_{i=1}^{M_k} \mathbb{I}_{l_p=k}}$

10:     **end for**

11:     Get covariance $C$ of $L$: $C \leftarrow \frac{1}{N} \sum_{k=1}^{N} \frac{\sum_{i=1}^{M_k} \mathbb{I}_{l_p=k}(f_{test,i} - \mu_k)(f_{test,i} - \mu_k)^{\mathrm{T}}}{\sum_{i=1}^{M_k} \mathbb{I}_{l_p=k}}$

12:     Space rotation: $\hat{\mu} \leftarrow \mathtt{SOBA}(\mu, C)$        ▷ See Equation equation 12 and equation 6

13:     **SOBA** logits: $logits_{trans} \leftarrow \mathtt{sim}(f_{test}, \hat{\mu})$

14:     Final inference: $logits \leftarrow logits_{clip} + \alpha \times logits_{\mathrm{trans}}$

15: **end for**

16: **return** $logits$                          ▷ return prediction based on the mode

---

---

**Algorithm 2** Queue update process

---

1: **Input:** CLIP logits of $f_{test}$: $logits_{ori}$, image embedding: $f_{test}$, pseudo-label of $f_{test}$: $l_p$, old queue: $L^{t-1}$, queue capacity: K.

2: **if** $|L_{l_p}^{t-1}| < K$ **then**

3:     $L_{l_p}^t \leftarrow \mathtt{EnQueue}(f_{test}, L_{l_p}^{t-1})$

4: **else**

5:     $\mathrm{H}_{max} \leftarrow \max(\mathrm{H}(L_{l_p}^{t-1}))$             ▷ Get the maximum entropy in $L_{l_p}^{t-1}$.

6:     **if** $\mathrm{H}(logits_{ori}) < \mathrm{H}_{max}$ **then**

7:         Dequeue feature with $\mathrm{H}_{max}$: $L_{l_p}^{t-1} \leftarrow \mathtt{DeQueue}(f_{test}^{ent}, L_{l_p}^{t-1})$

8:         Enqueue feature $f_{test}$: $L_{l_p}^t \leftarrow \mathtt{EnQueue}(f_{test}, L_{l_p}^{t-1})$

9:     **else**

10:         $L_{l_p}^t \leftarrow L_{l_p}^{t-1}$

11:     **end if**

12: **end if**

13: **return** $L^t$                              ▷ update the queue

---

