# OpenReview forum: "REORIENTING THE FROZEN SPACE: TRAINING-FREE TEST-TIME ADAPTATION BY GEOMETRIC TRANSFORMATION"
_ICLR.cc/2026/Conference — ICLR 2026 Conference Withdrawn Submission_

### Official Review · Reviewer_dkvg · 2025-10-22

**Soundness:** 2
**Presentation:** 2
**Contribution:** 2
**Rating:** 2
**Confidence:** 4

**Summary:**

The paper proposes to use orthogonal basis transformation on the embeddings in visual-language models (VLMs), such as CLIP, for better out-of-distribution (OOD) detection.

**Strengths:**

The paper studies VLMs, the frontier combining vision and language.

**Weaknesses:**

The method is poorly motivated, i.e., why did the authors select the orthogonal basis transformation? There is a comparison with other transformation methods; however, there is no evidence that orthogonal basis transformation must be the optimal transformation. For example, there are many more distances and transformations beyond those listed in Fig. 3, such as Minkowski distance, etc.

Importantly, it is pointed out that "standard basis is **suboptimal**" on page 4. However, can you prove that the orthogonal basis transformation used here is optimal? If the orthogonal basis transformation cannot be proved to be optimal, how do you know that other basis transformations are suboptimal?

**Questions:**

There is no theoretical proof showing the soundness of SOBA. For example, in Fig. 3(a), orthogonal basis transformation > Mahalanobis > NCM > Euclidean > Manhattan. But why? Does this hold true under every circumstance (settings, datasets, models, etc.)? If not (e.g., on some dataset in Table 2), why?

The application of the proposed method, SOBA, is too limited. The experiment is only about OOD. And SOBA cannot consistently achieve the best among the competitors. VLMs have a broad range of applications, not limited to OOD.

Is it possible to apply the method to some more recent, state-of-the-art VLMs, such as Qwen2.5-VL? If not, why?

Crucial: It is important to know where the improvement comes from. A simple number (higher/lower than SOTA by what percentage) cannot tell you much. What images are misclassified before using SOBA but corrected by SOBA? What images are well classified before using SOBA, but SOBA fails? Why? Are the corrected images truly because of the orthogonal basis transformation? I am not confident about the method without seeing this information about the in-depth analysis.

Moreover, is Figure 1 created using a real-world dataset, or simply a toy data distribution? If it is a toy distribution, please switch to the real data. You can show more (if not all) real-world distributions in the appendix. It is important to visually illustrate what this method does, geometrically, to the features.

---

### Official Review · Reviewer_QiLa · 2025-10-28

**Soundness:** 3
**Presentation:** 3
**Contribution:** 2
**Rating:** 4
**Confidence:** 4

**Summary:**

This paper proposes a training-free test-time adaptation (TTA) method for vision-language models, termed Space Reorienting with Basis Transformation (SOBA). The key idea is to reorient class prototypes into the subspace defined by the feature covariance matrix of test samples, thereby improving class separability in the feature space.
Specifically, SOBA maintains a dynamic queue of high-confidence test samples to estimate class prototypes. At test time, it performs eigenvalue decomposition of the feature covariance matrix and transforms the prototypes into the eigenspace of the input features. The final prediction combines the zero-shot prediction with that of the transformed prototype-based classifier.
SOBA is computationally efficient (requiring no backpropagation) and demonstrates strong performance under out-of-distribution (OOD) and cross-dataset benchmarks using CLIP models with transformer-based text encoders and either ResNet or ViT visual encoders.

**Strengths:**

* The proposed eigenspace-based prototype transformation is conceptually simple, computationally lightweight, and empirically effective across multiple OOD and cross-dataset benchmarks.
* The method fits within the training-free TTA category, requiring only forward-pass computations (specifically, eigenvalue decomposition), which makes it appealing for large-scale or latency-sensitive applications.

**Weaknesses:**

* Limited analysis of class separability effects: The core motivation of SOBA is to enhance inter-class separability—especially for hard classes with overlapping features—yet no ablation or quantitative analysis is provided to verify this. It remains unclear whether the transformation actually improves accuracy for previously hard-to-classify classes, or whether it introduces trade-offs that harm already well-separated classes. A deeper empirical study is needed to substantiate this motivation.
* Lack of justification for transforming only text embeddings: The transformation is applied solely to class prototypes derived from text embeddings, while image embeddings remain unchanged. Under distribution shift, image features may also vary non-uniformly across classes, potentially misaligning the transformed text space. The rationale for limiting the transformation to text embeddings should be clarified, along with possible benefits or drawbacks of extending it to image embeddings.
* Missing related work: The paper should cite Forward-Only Adaptation (FOA) (Niu et al., ICML 2024, Oral), which also achieves efficient TTA without backpropagation using forward-only adaptation. A conceptual or empirical comparison with FOA would strengthen the paper’s positioning within the training-free TTA literature.

**Questions:**

1. Can the authors analyze the effect of SOBA on class separability, e.g., by showing per-class accuracy improvements or changes in inter-class distances before and after transformation?
2. What is the reasoning behind transforming only the text embeddings rather than both modalities?
3. How does SOBA compare conceptually and empirically with FOA (Niu et al., 2024)?
4. Can the authors quantify the memory overhead of the dynamic queue? Since it scales linearly with the number of classes, this could be a concern for large-scale datasets like ImageNet.

---

### Official Review · Reviewer_QEvC · 2025-10-29

**Soundness:** 3
**Presentation:** 3
**Contribution:** 3
**Rating:** 6
**Confidence:** 5

**Summary:**

This paper introduces SOBA, a training-free test-time adaptation method for CLIP. Instead of modifying the feature extractor or updating parameters, the method estimates a global covariance structure from high-confidence test samples and uses principal components to define a new orthogonal basis. Class prototypes are then re-expressed in this basis, yielding re-oriented decision boundaries. The final prediction fuses the transformed similarity scores with the original CLIP logits.

**Strengths:**

1. The method adapts to test-time distribution shift through basis transformation of class prototypes, offering a clean and original idea.
2. The proposed method requires no fine-tuning or gradient updates, making it computationally efficient and deployment-friendly.
3. The algorithm is conceptually simple yet effective.
4. Evaluated across multiple datasets and distribution shift scenarios, showing consistent gains over zero-shot CLIP and several TTA baselines.
5. The use of shared covariance and principal components has a solid geometric motivation and is well explained.

**Weaknesses:**

1. Despite claims of general applicability, experiments are restricted to CLIP models (ViT-B/16, ResNet). The method’s effectiveness on other architectures (e.g., SigLIP series, other VLMs, CLIP with other visual backbones) remains untested.
2. The approach assumes that reorienting class prototypes suffices to adapt, but ignores potential distortions in the feature space itself under distribution shift.
3. The fusion weight $\alpha$ is tuned on ImageNet and fixed elsewhere. It is unclear if this choice generalizes across datasets or domains. It is also not fair as ImageNet is one of the test dataset.
4. Potential failure cases are not discussed.
5. Related work like TCA [1] on training-free TTA is notably absent in discussion and comparison.

[1] Is Less More? Exploring Token Condensation as Training-free Test-time Adaptation

**Questions:**

1. Did you observe cases where the feature extractor (e.g., CLIP encoder) itself outputs distorted features under shift, making prototype reorientation insufficient? Would adapting the encoder (e.g., via BN stats) help in such cases?
2. Your method updates the prototypes and basis every fixed percentage of test data. Since the test data is sequentially processed with random order (due to random seed), it is unclear why update frequency has such a strong effect, as shown in the FGVC dataset.

See weaknesses part.

---

### Official Review · Reviewer_Cfgx · 2025-10-29

**Soundness:** 3
**Presentation:** 3
**Contribution:** 3
**Rating:** 6
**Confidence:** 5

**Summary:**

This paper presents a training-free TTA method, SOBA, which addresses the limitation of class-boundary ambiguity commonly observed in static feature spaces under the training-free TTA paradigm. The core contribution lies in a simple visual prototype alignment strategy that sharpens decision boundaries by deriving orthogonal basis via singular value decomposition. Extensive experiments across 15 datasets demonstrate competitive performance and efficiency.

**Strengths:**

- The evaluation is comprehensive, including two experimental settings across 15 diverse datasets.
- The idea of updating queue prototypes through orthogonal bases is interesting.

**Weaknesses:**

- The paper lacks detailed justification and quantitative evidence regarding the improved decision boundaries. Specifically, the explicit relationship between the class matrix and the resulting decision boundaries needs to be clarified. Moreover, after transforming the feature space, how can one quantitatively verify that the decision boundaries are indeed better aligned or more discriminative?
- Another weakness is the lack of discussion regarding the most recent works, which may reduce the impact of this paper. For instance, works such as MPE [1] (which shares the same motivation of addressing fuzzy class-boundaries), TT-RAA [2] (a similar method involving covariance‐matrix estimation), BCA [3] (a faster approach that does not require maintaining a queue), and the very recent GS-Bias [4] and TCA [5] (2025 publications) are not sufficiently acknowledged or compared against.

[1] Multi-Cache enhanced Prototype Learning for Test-Time Generalization of Vision-Language Models. ICCV 2025

[2] Test-Time Retrieval-Augmented Adaptation for Vision-Language Models. ICCV 2025

[3] Bayesian Test-Time Adaptation for Vision-Language Models. CVPR 2025

[4] GS-Bias: Global-Spatial Bias Learner for Single-Image Test-Time Adaptation of Vision-Language Models. ICML 2025

[5] Is Less More? Exploring Token Condensation as Training-free Test-time Adaptation. ICCV 2025

**Questions:**

- It would be helpful to quantify the computational and memory overhead introduced by SVD, especially when scaling to larger models or higher-dimensional features.
- High-confidence erroneous samples in the queue may distort prototype updates and adversely impact the NCM-based classification performance. How do the authors consider addressing this over-confidence issue?
- Discussing failure cases of SOBA would be valuable for better understanding its limitations and the scenarios where the method may struggle.

---

### Official Review · Reviewer_T9Er · 2025-10-31

**Soundness:** 2
**Presentation:** 3
**Contribution:** 2
**Rating:** 2
**Confidence:** 4

**Summary:**

This paper proposes SOBA, a training-free, test-time adaptation (TTA) method for Vision-Language Models. The authors identify that existing training-free methods are limited by the fixed geometry of the pre-trained feature space, which can lead to poor separability for semantically similar classes. SOBA's approach is to (1) maintain a dynamic queue of high-confidence, pseudo-labeled test samples, (2) compute a shared covariance matrix from these samples and use SVD (PCA) to find a new orthogonal basis, and (3) re-express the class prototypes (class means) in this new, test-aligned basis. The final prediction is an ensemble of the original zero-shot classifier and a new classifier (NCM) that compares test features to these reoriented prototypes.

**Strengths:**

The paper identifies a valid and important problem in training-free TTA: the fixed, pre-trained feature space can be suboptimal for a specific test distribution, especially when classes overlap. The core idea of finding a training-free way to adjust the decision boundaries to better fit the test data is a well-motivated and practical research direction. The method itself is simple, intuitive, and computationally lightweight, as it primarily relies on SVD and vector-matrix multiplication rather than expensive backpropagation.

**Weaknesses:**

1. The paper frames the method as a novel "geometric transformation" that "reorients" and "reshapes" the feature space. However, the core methodology (Sec. 3.3) is a standard, well-known statistical technique. The method computes a shared covariance matrix $C$ from test-set features and finds its principal components $Q_c$ via SVD. It then transforms the class prototypes $\mu$ using this matrix ($\hat{\mu} = \mu Q_c$). This is not a new form of geometric transformation; it is a standard implementation of a Nearest Centroid (NCM) classifier in a space transformed by Principal Component Analysis (PCA). The paper's own baseline comparisons to "PCA*" and "LDA*" (variants constructed by the authors) confirm that the proposed method is, at best, a very specific and incremental implementation of these classical techniques, not a new paradigm. The justification in Appendix D ("WHY SOBA IS NOT SIMPLE PCA?") is weak, arguing that SOBA only transforms the prototypes, not the features, which is an implementational choice, not a fundamental methodological innovation.

2. The method's efficiency and simplicity rely on a strong "GDA assumption" (Line 248) of a shared covariance matrix $C$ for all classes. This is a very restrictive assumption that is unlikely to hold true in complex, real-world OOD distributions. The paper brushes this aside in Table 4 by claiming it has "little impact on... performance" but "significantly reduces inference time." This trade-off is not properly analyzed and feels more like a post-hoc justification for a simplifying assumption.

3. For a paper claiming state-of-the-art in training-free TTA, the experimental comparison is incomplete. It omits a significant number of recent and directly relevant training-free and transductive methods that also operate on VLM feature spaces. Key missing baselines include (but are not limited to) [1-4]. A convincing claim of SOTA would require comparison against these contemporary works.

4. The entire method hinges on building a dynamic queue of high-confidence samples to estimate the covariance matrix. This creates a potential failure case: in scenarios with significant domain shifts or high-entropy datasets (like Aircraft), the model's initial predictions may be systematically wrong, yet still "high-confidence." This would lead to a queue populated with incorrect samples, resulting in a corrupted basis transformation that degrades performance rather than improving it. The paper's robustness analysis (Appendix C.1.4) only injects random noise and does not adequately address this more realistic failure mode.



[1] Ra-tta: Retrieval-augmented test-time adaptation for vision-language models, ICLR2025

[2] Bayesian test-time adaptation for vision-language models, CVPR2025

[3] Awt: Transferring vision-language models via augmentation, weighting, and transportation, NeurIPS2024

[4] Efficient and context-aware label propagation for zero-/few-shot training-free adaptation of vision-language model, ICLR2025

**Questions:**

1. Can the authors clearly differentiate the proposed method from a standard NCM classifier applied in a PCA-transformed space? The justification in Appendix D seems to be the only difference, and it's an implementational choice, not a fundamental methodological leap.

2. How does the method perform if the GDA assumption is violated, for example, on a benchmark known to have classes with highly distinct covariance structures?

---

### Note · Authors · 2025-11-12

I have read and agree with the venue's withdrawal policy on behalf of myself and my co-authors.